# Metastatic Lymph Node Detection on Ultrasound Images Using YOLOv7 in Patients with Head and Neck Squamous Cell Carcinoma

**DOI:** 10.3390/cancers16020274

**Published:** 2024-01-08

**Authors:** Sato Eida, Motoki Fukuda, Ikuo Katayama, Yukinori Takagi, Miho Sasaki, Hiroki Mori, Maki Kawakami, Tatsuyoshi Nishino, Yoshiko Ariji, Misa Sumi

**Affiliations:** 1Department of Radiology and Biomedical Informatics, Nagasaki University Graduate School of Biomedical Sciences, 1-7-1 Sakamoto, Nagasaki 852-8588, Japan; sato@nagasaki-u.ac.jp (S.E.); katt@nagasaki-u.ac.jp (I.K.); yuki@nagasaki-u.ac.jp (Y.T.); sasaki-m@nagasaki-u.ac.jp (M.S.); morih@nagasaki-u.ac.jp (H.M.); m-kawa03@nagasaki-u.ac.jp (M.K.); bb25119037@ms.nagasaki-u.ac.jp (T.N.); 2Department of Oral Radiology, Osaka Dental University, 1-5-17 Otemae, Chuo-ku, Osaka 540-0008, Japan; fukuda-m@cc.osaka-dent.ac.jp (M.F.); ariji-y@cc.osaka-dent.ac.jp (Y.A.)

**Keywords:** metastatic lymph node, squamous cell carcinoma, ultrasonography, YOLOv7, deep learning, computer-assisted diagnosis

## Abstract

**Simple Summary:**

Cervical lymph node (LN) metastasis is a critical prognostic factor for patients with head and neck squamous cell carcinoma (HNSCC), rendering accurate diagnosis of LN metastasis crucial for improving patient outcomes. Our study aimed to develop deep learning models for metastatic LN detection using YOLOv7, the fastest single-stage object detection model, on B-mode and power Doppler (D-mode) ultrasonography in patients with HNSCC and investigate their utility in supporting the diagnosis by comparing their performance to that of highly experienced radiologists and less experienced residents. A total of 462 B- and D-mode ultrasound images were used to train, validate, and test the B- and D-mode models, respectively. The detection performances of the B- and D-mode models for metastatic LNs were higher than those of less experienced residents; the performance of the D-mode model was comparable to that of highly experienced radiologists, suggesting that YOLOv7-based models are useful for supporting the diagnosis.

**Abstract:**

Ultrasonography is the preferred modality for detailed evaluation of enlarged lymph nodes (LNs) identified on computed tomography and/or magnetic resonance imaging, owing to its high spatial resolution. However, the diagnostic performance of ultrasonography depends on the examiner’s expertise. To support the ultrasonographic diagnosis, we developed YOLOv7-based deep learning models for metastatic LN detection on ultrasonography and compared their detection performance with that of highly experienced radiologists and less experienced residents. We enrolled 462 B- and D-mode ultrasound images of 261 metastatic and 279 non-metastatic histopathologically confirmed LNs from 126 patients with head and neck squamous cell carcinoma. The YOLOv7-based B- and D-mode models were optimized using B- and D-mode training and validation images and their detection performance for metastatic LNs was evaluated using B- and D-mode testing images, respectively. The D-mode model’s performance was comparable to that of radiologists and superior to that of residents’ reading of D-mode images, whereas the B-mode model’s performance was higher than that of residents but lower than that of radiologists on B-mode images. Thus, YOLOv7-based B- and D-mode models can assist less experienced residents in ultrasonographic diagnoses. The D-mode model could raise the diagnostic performance of residents to the same level as experienced radiologists.

## 1. Introduction

Cervical lymph node (LN) metastasis is one of the most relevant prognostic factors for patients with head and neck squamous cell carcinoma (HNSCC) [1,2,3,4]. Therefore, it is important to detect and treat metastatic LNs as early as possible to improve patients’ prognosis. Computed tomography (CT), magnetic resonance imaging (MRI), ultrasonography (US), and positron emission tomography/CT (PET/CT) are used to detect metastatic LNs [5,6,7,8]. As each modality has advantages and disadvantages, a multimodal assessment can yield an effective imaging diagnosis of metastatic LNs. Contrast-enhanced CT and contrast-enhanced MRI are the most commonly used due to their superior anatomical resolution [5,9,10,11]. They are useful for detecting LNs with internal necrosis characteristic of lymph nodes in HNSCC [10]. In particular, the high contrast resolution of MR makes it easier to detect internal abnormalities than CT [5]; however, the long scan time of MRI can produce motion artifacts, which impact LN evaluation. In contrast, CT has a shorter scan time and is the most useful modality for surveying large LNs; however, it has the disadvantage of ionizing radiation exposure. PET/CT also involves exposure to ionizing radiation. Although Sun et al. reported that the sensitivity and specificity of ^18^FDG-PET/CT were 0.84 and 0.96, respectively, for detecting regional LN metastasis [12], PET/CT is still complementary to the conventional radiological investigations [13,14]. Recently, PET/MRI has been reported to have better diagnostic ability for LN metastasis than contrast-enhanced CT [1,15,16]; however, PET/MRI has not been widely used.

US is a simple, ionizing-radiation-free, non-invasive, and low-cost imaging modality; owing to its higher spatial resolution, it can facilitate better detection of intranodal architectural changes compared with CT and MRI [5,6,7,17]. Although US cannot evaluate deep LNs, such as the retropharyngeal nodes, it is the first-line modality for the intensive evaluation of relatively superficial LNs that appear enlarged on CT and/or MRI in patients with HNSCC [7,18,19,20].

The US diagnosis of metastatic LNs is usually made using gray-scale (B-mode) US, which provides morphological information on the size, shape, borders, echotexture, and power, or color Doppler US, which provides vascular information [5,18]. Metastatic LNs are reportedly associated with absent hilar echoes, increase in the short-axis length, near circular form, and heterogeneous internal echoes in B-mode images; and compressed hilar flow, peripheral flow, and/or scattered parenchymal flow signals in power or color Doppler US [5,7,20]. Power Doppler US is more sensitive than color Doppler US for visualizing smaller blood vessels because its greater dynamic range enhances the visibility of microvascularity [5,21]. However, US is examiner-dependent, i.e., accurate diagnosis depends on the operator’s expertise [6,19,22,23], making accurate diagnosis a challenge for inexperienced examiners. Furthermore, experienced radiologists may face difficulty maintaining high diagnostic performance for metastatic LNs due to the diagnostic work overload. This leads to patients missing appropriate treatment. To address this issue, implementing a computer-assisted diagnosis (CAD) system using deep learning (DL) that automatically identifies metastatic LNs can benefit inexperienced and experienced radiologists by compensating for inexperience and preventing mistakes.

Recently, DL applications have been reported for the diagnosis of metastatic LNs in patients with HNSCC [11,24,25,26,27]. Santer et al. reported that the mean diagnostic accuracy of artificial intelligence for LN classification was 86% (range: 43–99%) [25], and Ariji et al. reported that metastases were diagnosed more accurately in the segmented lymph nodes in contrast-enhanced CT images using the DL model compared with the radiologists’ interpretation [26]. However, most previous studies have used CT and PET-CT images for DL [25].

Currently, DL using US images is being used for the diagnosis of breast lesions [28,29,30,31], submandibular gland inflammation [32], Sjögren’s syndrome [33], and thyroid nodules [34,35]. The application of DL to the US diagnosis of LNs has been reported for axillary LNs (accuracy = 72.6%) [36] and metastatic LNs from thyroid cancer (accuracy = 83%) [37]. Recently, Zhu et al. reported that a DL radiomics model based on B-mode and color Doppler US images showed better diagnostic performance than skilled radiologists for four common etiologies (metastatic, lymphoma, tuberculous, and reactive) of unexplained cervical lymphadenopathy [19]. However, few DL models have been devised for metastatic LN detection using US images in patients with HNSCC. Thus, we aimed to develop DL models for CAD for the US diagnosis of cervical metastatic LNs in patients with HNSCC.

The following CNN models were utilized for the LN metastasis diagnosis using CT in HNSCC: ResNet, DIGITS, BoxNet, SmallNet, DualNet [25], U-net [26], Xception [24], AlexNet [27], DetectNet [11], and DualNet [38]. The AUCs for diagnosing metastatic LNs were as follows: 0.91 using DualNet [38], 0.8 using AlexNet [27], 0.95 using U-net [26], 0.898 and 0.967 at level I–II and level II, respectively, using Xception [24]. We selected the You Only Look Once (YOLO) real-time multiple object detector, which has high efficiency and remarkable speed of operation [39], as the ideal algorithm for US diagnostic models. This is because, in actual US examinations, US videos are used for assessment, and YOLO allows real-time multiple detection of metastatic LNs in live US video examinations. While this study employed static images as a preliminary step in the development of the DL model for US diagnosis, our ultimate goal is to create a DL model capable of real-time detection of metastatic LNs during live US video examinations. In this study, the YOLO version 7 (YOLOv7) algorithm was used since previous studies have suggested that YOLOv7 provides greater accuracy and requires less computation time [40,41,42,43,44,45]. The salient features of YOLOv7 used in this study are as follows:Incorporates a trainable bag-of-freebies to improve real-time object detection performance without increasing inference costs;Integrates extended and composite scaling to effectively reduce model parameters and calculations for faster detection; andProvides predesigned freebies to facilitate model fine-tuning and simplifies the addition of modules and the creation of new models, which are characterized by higher detection accuracy, speed, and convenience [40,41,42,43,44,45].

Therefore, this study aimed to develop YOLOv7-based models using B-mode images and D-mode images (B-mode superimposed power Doppler images) for detecting metastatic LNs and investigate the utility of the models for CAD by comparing the detection performance for metastatic LNs with those of highly experienced radiologists and less experienced residents.

## 2. Materials and Methods

### 2.1. Patients

This study was approved by the Institutional Review Board (IRB) of Nagasaki University Hospital (No. 11072593-13). The IRB waived the requirement for informed consent from the participants owing to the retrospective study design. The study protocol conformed with the ethical guidelines of the Declaration of Helsinki and the Ethical Guidelines for Medical and Health Research involving Human Subjects by the Ministry of Health, Labor, and Welfare of Japan.

A radiologist (S.E.) with 26 years of experience in neck US diagnosis selected the ultrasonographic images of LNs from our hospital imaging database of patients who underwent neck US examination for the assessment of nodal metastasis from HNSCC and neck dissections between October 2008 and January 2022.

The inclusion criteria for the LNs used in this study were as follows:Both axial-oriented B- and D-mode US images were available;Short-axis diameter >2 mm (the longest nodal axis perpendicular to the long axis of the node with the maximal nodal area in an axially oriented US image);Identifiable on dissection specimens; andHistologically proven metastasis or non-metastasis.

The identification of the dissected nodes and US images was facilitated by LN schema maps that recorded the location of US-scanned LNs relative to the surrounding anatomic structures, such as vessels, salivary glands, bones, and muscles, as reference. LNs with unclear images due to artifacts or incomplete operations were excluded.

Consequently, 540 LNs (261 metastatic and 279 non-metastatic LNs) of 126 patients with HNSCC, including 78 men and 48 women with a mean age of 63 years (range, 31–91 years), were enrolled in this study, and one B-mode and one D-mode image of each LN were used. Since 2–3 LNs were often observed in one US image, a total of 462 B-mode and 462 D-mode images for 540 LNs were prepared.

The number of LNs at each cervical level was 16, 203, 210, 70, 29, and 12 for levels IA, IB, II, III, IV, and V, respectively. The primary sites of SCC nodes were the tongue (*n* = 49), gingiva (*n* = 52), oropharynx (*n* = 8), buccal mucosa (*n* = 5), maxillary sinus (*n* = 1), maxilla (*n* = 1), palate (*n* = 1), oral floor (*n* = 6), and lip (*n* = 3).

### 2.2. US Image Acquisition

US examinations were performed using a 9-unit LOGIQ (GE Healthcare, Milwaukee, WI, USA) equipped with a wide-bandwidth (range, 9–14 MHz) transducer by three radiologists with 26 years’ or more experience in neck US. B-mode US was performed at a frequency of 14 MHz. D-mode US was performed at 6.6 MHz, and standardized power Doppler settings were chosen to optimize the detection of vascular signals in and around the LNs, which had low-velocity flow. Common settings of 900 Hz pulse repetition frequency and a wall filter of 133 Hz were used. Both B- and D-mode images of the largest diameter on the transverse orientation of each detected LN were saved; the B-mode and D-mode images of each LN were almost identical in positioning. The area set by each radiologist to detect the Doppler signal of the LNs during the US examination was bounded by a square line around the LNs (indicating areas containing metastatic and non-metastatic LNs) only in the D-mode image.

### 2.3. Preparation of US Datasets

The 462 B-mode and 462 D-mode images (373 × 393 pixels, 3.8 × 4 cm) of 540 LNs were downloaded from our imaging database, and the deepest area (373 × 20 pixels, 3.8 × 0.2 cm) with almost no echo signal from each image was cropped by a radiologist with 26 years’ experience in neck US diagnosis. Finally, all images were resized to 640 × 640 pixels and saved in the PNG format for input into YOLOv7 (Figure 1) [44].

The B- and D-mode images were assigned to the training (*n* = 324), validation (*n* = 92), and testing datasets (*n* = 46) so that the distribution of images was approximately 7:2:1 (Table 1 and Figure 1). None of the images of patients assigned to the testing dataset crossed over to the training or validation datasets.

### 2.4. YOLOv7 Model Procedure

Metastatic LN detection models were created using the YOLOv7 network [31] on a Windows 10 Pro (Microsoft) desktop computer with a 24 GB GPU (NVIDIA GeForce RTX 3090, Santa Clara, CA, USA). The YOLOv7 network was downloaded from the GitHub repository “https://github.com/WongKinYiu/yolov7” (accessed on 21 April 2023).

YOLOv7 was separately trained for the detection of metastatic LNs on B-mode and D-mode images to create optimized models for each mode (B-mode and D-mode models, respectively) using 324 images for training and 92 images for validation; the bounding boxes indicated metastatic LNs with annotation labels specifying the coordinates. These annotations were generated using the image data annotation software “LabelImg version 1.8.1” (accessed on 8 March 2023) by a radiologist (S.E.) with 26 years of experience in neck US diagnosis. She annotated each LN using a bounding box to encompass the entire LN without overlapping its margins by comparing the LN schema maps with the pathology findings. Subsequently, another radiologist (M.SU.) with 27 years of experience in neck US diagnosis confirmed the accuracy and consistency of all images.

The network architecture of YOLOv7 consists of three main components: backbone, neck, and prediction (Figure 1) [44,45]. The backbone incorporates CBS layers, which consist of convolution, batch normalization, and SiLU activation functions; max pooling layers; and efficient layer aggregation network (ELAN) layers [44,45]. In the neck, the SPPCSPC layer is created by integrating the spatial pyramid pooling (SPP) and convolutional spatial pyramid (CSP) architectures. The feature map output of the SPPCSPC layer is divided into two parts, which enhances the perceptual range of the network [44]. The CUC layer serves as the basic unit for combining feature maps, involving convolution, up-sampling, and combining feature maps. The REP layer is an innovative concept that uses structural reparameterization to modify the structure during inference to improve the model’s performance. The prediction layer of YOLOv7 generates feature maps of three different sizes during the prediction process [45]. In our study, the architecture of YOLOv7 remained unchanged, and the initial learning rate was set at 0.01 and 1000 epochs for training.

YOLOv7 presents multiple bounding boxes, their class labels, and confidence scores [41]. The confidence score is calculated as the product of the predicted value (object) and the intersection over the union of the bounding box around the object. This score serves as a measure of how well a given bounding box matches the actual object, with higher confidence scores indicating greater accuracy of detection. In this study, the B- and D-mode models, which were optimized for metastatic LNs in the B- and D-mode images, respectively, displayed boxes surrounding the objects determined to be metastatic LNs and their confidence scores for the test images. While evaluating the detection performance of the B- and D-mode models, the following types of thresholds were established for the confidence score:A low threshold (confidence score ≥ 0.1) to obtain a higher recall (B-mode model-1 and D-mode model-1), andAn investigated threshold to obtain the largest area under the receiver operating characteristic curve (AUC) for the test images (B-mode model-2 and D-mode model-2).

### 2.5. Evaluation of Detection Performance and Comparison with Observers

The detection performance of the metastatic LNs was evaluated using recall, precision, F1-scores, false-positive rate for non-LN areas, accuracy (Table 2), and AUC. The false-positive rate for non-LN areas was calculated as the number of non-LN areas incorrectly detected as metastatic LNs in one test image (number of test images = 46) [11]. The accuracy and AUC were evaluated for all LNs for the test images, under the assumption that the undetected LNs were non-metastatic LNs. Additionally, the recall for the metastatic LNs at each cervical level (levels I, II, and III + IV) was examined.

Furthermore, two highly experienced radiologists (M.SA. and Y.T., with 27 years of experience in neck US diagnosis) and two less experienced dental residents (H.M. and M.K.) evaluated the same testing datasets after calibration. Calibration was performed using 20 randomly selected images (the first 10 metastatic and 10 non-metastatic LN images numbered within each of the metastatic and non-metastatic LN groups from the B- and D-mode training datasets). Although the selected images represented typical metastatic and non-metastatic LNs, they may not have been representative of the entire dataset. However, they were sufficient for understanding the typical diagnostic criteria for metastatic and non-metastatic LNs using B- and D-mode images.

The four observers were blinded to the patient’s clinical information and instructed to interpret the test images after calibration. There was an interval of more than one week between the B-mode- and D-mode image evaluations. Thereafter, the performances of the models were compared with those of the observers.

The interobserver agreement between the two highly experienced radiologists for the identification of metastatic LNs in the B- and D-mode images was substantial and almost perfect (Cohen’s weighted kappa coefficient: 0.894 and 0.682 for the B-mode and D-mode images, respectively). Therefore, disagreements on imaging interpretation between the two radiologists were resolved by discussion, and the evaluation results of the two radiologists were combined into a single finding for use in the subsequent analysis. The two residents, unlike the radiologists, were less experienced; therefore, all test cases were evaluated consensually by the two residents, culminating in a single evaluation result.

### 2.6. Statistical Analysis

The optimal thresholds for the confidence score at which the B- and D-mode models had the largest AUCs, respectively, were obtained using the Youden index. The recalls for metastatic LNs and accuracies for all LNs in the test images were compared between the models and observers using the binominal test. The AUCs of the diagnostic performance for all LNs in the test images were compared between the models and three observers using the DeLong test [46]. Statistical significance was set at *p* < 0.05. Agreements on the identification of metastatic LNs between the models and observers were assessed using Cohen’s weighted kappa coefficient. The kappa coefficients were interpreted as follows: 0–0.2, poor agreement; >0.2 and ≤0.4, fair agreement; >0.4 and ≤0.6, moderate agreement; >0.6 and ≤0.8, substantial agreement; and >0.8 and ≤1.0, almost perfect agreement. Statistical analyses were performed using JMP Pro (version 17.0.0; SAS Institute Inc., Cary, NC, USA) and IBM SPSS, version 27.0.1 (IBM, Armonk, NY, USA).

## 3. Results

### 3.1. Detection Performance for Metastatic LNs

The detection results for metastatic LNs in the B- and D-mode images using the B- and D-mode models, respectively, are shown in Figure 2. The optimal thresholds for the confidence score at which the B- and D-mode models had the largest AUCs were 0.23 and 0.48, respectively (Figure 3). The recall, precision, F1-score for metastatic LNs, false-positive rate per non-LN area, accuracy, and AUC of B- and D-mode model-1 (confidence score of 0.1 for both) and B- and D-mode model-2 (confidence score of 0.23 and 0.48, respectively), highly experienced radiologists, and less experienced residents are shown in Table 3 and Figure 3.

For the B-mode images, B-mode model-1 and model-2 had the same recall value of 0.75; however, the other evaluation metrics besides recall were higher for B-mode model-2 than those for B-mode model-1. All evaluation metrics of B-mode model-1 and model-2 were lower than those of the radiologists, and the accuracy and AUC of B-mode model-1 were significantly lower than those of the radiologists (*p* = 0.031 and 0.0166, respectively). Moreover, all evaluation metrics of B-mode model-2 and evaluation metrics other than the accuracy and AUC of B-mode model-1 were higher than those of the residents; however, the differences in recall, accuracy, and AUC between B-mode model-1 and model-2 and the residents were not significant. The recall, accuracy, and AUC of the radiologists were significantly higher than those of the residents (*p* = 0.008, 0.012, and 0.0022, respectively) (Table 4).

For the D-mode images, the recall values of D-mode model-1 and model-2 for metastatic LNs were 0.821 and 0.75, respectively. The recall of D-mode model-1 was not only higher than that of the residents (0.714) but also that of the radiologists (0.75), although the difference lacked statistical significance (Table 4). However, the evaluation metrics other than recall were higher in D-mode model-2 than those in D-mode model-1, and all evaluation metrics of D-mode model-2 (recall = 0.75, precision = 0.84, F1-score = 0.792, false-positive rate for non-LN area = 0, accuracy = 0.814, and AUC = 0.81) were comparable to those of the radiologists. All evaluation metrics of D-mode model-1 and model-2 were higher than those of the residents, although there were no statistically significant differences in recall, accuracy, or AUC between them (Table 4).

D-mode model-1 and model-2 showed higher precision, F1-score, accuracy, and AUC than those of B-mode model-1 and model-2, respectively, although the differences in recall, accuracy, and AUC were not significant. Additionally, the false-positive rates for non-LN areas in D-mode model-1 and model-2 (0.022 and 0, respectively) were lower than those of B-mode model-1 and model-2 (0.087 and 0.043, respectively). B-mode model-1 had three false positives for the cross-section of the sternocleidomastoid muscle and one false positive for the internal jugular vein, while D-mode model-1 had one false positive for the cross-section of the internal jugular vein (Figure 2a).

All evaluation metrics of the radiologists were higher for the B-mode images than the D-mode images, and all values other than the accuracy and AUC of the residents were higher for the D-mode images than the B-mode images, although there was no statistically significant difference in recall, accuracy, and AUC between them (Table 4).

### 3.2. Recall at Each Cervical Level

The recall for metastatic LNs at each cervical level is shown in Table 5. For the B-mode images, recalls at level I of B-mode model-1 and -2 and the radiologists had the same high values of 0.909, which were significantly greater than those of the residents. However, the recalls of B-mode model-1 and model-2 at level III + IV (0.429 for both) were lower than those of the radiologists and residents. For the D-mode images, the recall of D-mode model-1 and the radiologists was the same for levels I and II (0.727 and 1, respectively); however, the recall of D-mode model-1 (0.714) for level III + IV was higher than that of the radiologists and residents.

### 3.3. Agreement on the Identification of Metastatic LNs between the Models and Observers

The agreement of identification for metastatic LNs between the B-mode models and the radiologists’ reading of B-mode images was fair and moderate (kappa values = 0.392 and 0.483 for B-mode model-1 and model-2, respectively), and those between the D-mode models and the radiologists for D-mode images were moderate (kappa values = 0.496 and 0.546 for B-mode model-1 and model-2, respectively) (Table 6).

## 4. Discussion

This study devised YOLOv7-based B- and D-mode models to detect metastatic LNs in B- and D-mode US images, respectively. To the best of our knowledge, this is the first study to develop DL models for the detection of metastatic LNs on US in patients with HNSCC. In addition, this study is novel in that it implemented the fastest and most accurate real-time object detection algorithm, YOLOv7. Further, this study provides valuable information regarding the performance of B- and D-mode models under the following types of confidence score thresholds: (1) a low threshold to obtain a higher recall (0.1 for both B- and D-mode model-1), and (2) a higher threshold to obtain the largest AUC (0.23 and 0.48 for B- and D-mode model-2, respectively), resulting in D-mode model-2 having a trade-off of decreasing not only false positives but also true positives.

In our study, the recall of D-mode model-1 for metastatic LNs was 0.821, which was the highest among the B- and D-mode models and higher than that of the experienced radiologists’ (0.75) and residents’ (0.714) reading of D-mode images, suggesting that D-mode model-1 can enhance the detection performance of metastatic LNs for not only residents but also experienced radiologists in D-mode images. In addition, the D-mode model-2 with a higher confidence score threshold had a recall of 0.75, precision of 0.84, F1-score of 0.79, accuracy of 0.814, and AUC of 0.81, which means that all the evaluation metrics (except recall) were higher than those of D-mode model-1, owing to fewer false positives, which was comparable to the reading by radiologists with 27 years’ experience and higher than the residents’ reading of D-mode images. Therefore, D-mode model-2 is a useful CAD system to raise the detection performance of residents to the same level as that of experienced radiologists for D-mode images. Furthermore, the recall of B-mode model-1 and model-2 for metastatic LNs was 0.75, which was lower than that of the radiologists but higher than that of the residents for B-mode images, suggesting that B-mode models are beneficial for improving the detection rate of metastatic LNs by less experienced residents.

Compared with radiologists, the B- and D-mode models had lower precision owing to the greater rate of false positivity. B-mode model-1 had four false positives for non-LN structures in the B-mode test images, leading to poor detection performance. In the D-mode images, blood vessels can be distinguished from other tissues based on blood flow signals, which is not possible in B-mode images; thus, less experienced residents found many false positives by misinterpreting blood vessels as metastatic LNs (12 false positives). In contrast, the experienced radiologists had no false positives for non-LN structures in either the B- or D-mode images, probably because they were completely familiar with the neck anatomy and had no difficulty distinguishing LNs from blood vessels and other structures. Therefore, the B-mode model, which was only trained on B-mode images, was also assumed to have many false positives for blood vessels; however, contrary to this assumption, B-mode model-1 had only one false positive for the internal jugular vein, and the other three false positives were for cross-sections of the sternocleidomastoid muscle (Figure 2a). Thus, we opine that the B-mode model could distinguish between metastatic LNs and blood vessels based on features other than blood flow signals (size, internal signals, location, etc.), but that learning the present image data was not sufficient to distinguish metastatic LNs from the muscle. Training with more B-mode image data would help in learning neck anatomy, such as muscle location, and decrease the false positive rate for the non-LN area.

D-mode model-1 had only one false positive case for the non-LN area, most likely because of the following advantages over B-mode models: (1) LNs could be distinguished from blood vessels and muscles by the Doppler blood flow signal, and (2) the area delineated by the radiologist during the US examination for the investigation of blood flow findings in detected LNs was bounded by a square in each D-mode image. Therefore, the D-mode model might have prioritized the detection of metastatic LNs within the square’s boundaries. Only one false positive for the non-LN area by D-mode model-1 was registered for the internal jugular vein, which spanned both the inside and outside of the square and was centrally located on the outside, as shown in Figure 2a, suggesting that the D-mode model may have been learning not only the inside but also the outside of the square. However, this was the only instance in which metastasis was registered outside of the square. Therefore, if the entire image, not only the LN area, is bounded by a square, the detection performance might differ.

However, unlike inexperienced residents, experienced radiologists usually do not misdiagnose non-LNs as LNs, as observed in our study. It is more difficult to identify whether the detected LN is metastatic; thus, they determine whether the LN detected in the B-mode image is metastatic by adding the blood flow information in the D-mode image. Therefore, the D-mode model trained on D-mode images with square-shaped bounded LN areas was similar to the actual evaluation method of radiologists; D-mode model-1 showed a higher recall than that of radiologists, suggesting that the D-mode model is beneficial for radiologists.

The higher detection performance of radiologists’ reading of B-mode images than on D-mode images is consistent with a study by Chikui et al., who reported that vascular information in D-mode images offered fewer predictive advantages for experienced radiologists in the diagnosis of metastatic LNs, whereas it can assist less experienced observers in distinguishing between LNs and blood vessels [23]. However, the recall of D-mode model-1 was higher than that of the radiologists. The kappa values between D-mode model-1 and radiologists were not high (0.496), suggesting that the D-mode models may have employed more accurate vascular information criteria than those used by the radiologists for the diagnosis of metastatic LNs. An improved D-mode model trained with more data is expected to further improve the detection performance of metastatic LNs.

This study had some limitations. First, the number of patients and LNs was small because this was a single-center study that used only US images acquired with the same apparatus and imaging parameters. However, especially at level III + IV, which had the fewest LN images, the recalls of the B- and D-mode models were lower than those of the other levels. Since the detection performance is related to the amount of training data, the performance can be improved by using more data. In addition, multicenter studies should be conducted to accumulate more data, which would allow for the construction of models with better generalization performance. Further, supplementing real-life data with generative adversarial network (GAN) [47,48] and computational fluid dynamics (CFD) simulations [49,50] can effectively expand the dataset.

Second, we did not conduct a model interpretability analysis such as Grad-CAM [51,52]; therefore, the basis for the model’s decision is unknown. The high kappa values of 0.894 and 0.682 for the B- and D-mode images between the two radiologists suggest that they used similar diagnostic criteria, whereas the lower kappa values between the models and the radiologists suggest that the models used different diagnostic criteria from the radiologists. The low kappa values between the B-mode models and the radiologists (0.392 and 0.483 for B-mode model-1 and -2, respectively) suggest that the B-mode model failed to learn the various subtle features that distinguish between metastasis and non-metastasis, which resulted in lower performance of B-mode models; experienced radiologists have learned this over the years as diagnostic criteria. In particular, the precision of radiologists was 0.958 for B-mode images, which was much higher than those of the B-mode models, suggesting that radiologists have fewer false positives and better criteria for identifying non-metastatic LNs than B-mode models. Possible reasons are as follows: the radiologists are more experienced than our study’s dataset in diagnosing metastatic and non-metastatic LN images, and they are familiar with the various US features of metastatic and non-metastatic LNs; since the number of non-metastatic LNs is much larger than that of metastatic LNs, radiologists may be particularly sensitive to the various features of non-metastatic LNs. In contrast, the dataset in this study was too small for the model to learn the variety of metastatic and non-metastatic US features and was insufficient for the generalizability of non-metastatic LN images. For example, the B-mode model may have mainly used size and rounded morphology to determine LN metastasis. However, since our study did not conduct a model interpretability analysis, the basis for the model’s diagnosis could not be confirmed. To enhance the reliability of the models, it is essential to clarify the basis for the model’s diagnosis using techniques such as Grad-CAM and confirm the differences in the diagnostic criteria between the radiologists and the models. In addition to learning with more data, insights into the basis of the model’s diagnosis can improve overall performance and reveal its strengths and limitations.

Third, the US images used in this study did not include images of neck masses other than LNs, such as thyroglossal duct cysts [53], branchial cleft cysts, lymphangiomas, or salivary gland tumors [54]. In clinical practice, some patients may have neck masses other than LNs. It should be noted that our models may produce false-positive results for neck masses other than metastatic LNs. Therefore, at this stage, the application of our models would be limited to screening; however, it would still aid the inexperienced examiner in diagnosis.

Fourth, distinct models were created in this study using the B- and D-mode images separately. However, in actual US examination, the radiologist combines the findings of both the B- and D-mode images for diagnosis. Zhu et al. developed a fusion model based on B-mode and D-mode images and achieved higher diagnostic performance for unexplained cervical lymphadenopathy than radiologists [19]. Therefore, the development of models to evaluate combined B- and D-mode images is expected to improve the detection performance of metastatic LNs in patients with HNSCC.

Fifth, since this was a retrospective study, only the largest static transverse section of each LN saved by a radiologist during the examination was used. Recently, it was reported that the model performance is superior with ultrasound videos compared to static ultrasound images [22]. YOLOv7, used in our study, and the latest version, YOLOv8 [55], are the fastest and most accurate real-time object detection models, which are suitable for multiple object detection from video files in real time. The development of a YOLO-based model that can detect metastatic LNs in real time during US examinations will be one of the most important themes of future research, necessitating multicenter studies with vast quantities of US video data.

## 5. Conclusions

The results of this study suggest that the B- and D-mode models based on YOLOv7 are valuable tools in supporting the US detection of metastatic LN in patients with HNSCC, particularly the D-mode model, which could improve the diagnostic performance of less experienced residents to the same level as experienced radiologists. Further performance improvements may be achieved by using more US data from multicenter studies and insights into the basis of the model’s diagnosis using interpretability analysis. Notably, the diagnostic accuracy of the radiologists in the actual US examination is expected to be higher than the results of this study, as radiologists also consider other relevant clinical findings, such as the location of the primary tumor and the detected LN, which are important clues for the diagnosis of metastatic LNs. Therefore, a CAD system using a multimodal model incorporating CT and/or MRI scans of the primary tumor and other clinical findings would be more practical and perform best in metastatic LN diagnosis with patients with HNSCC.

## Figures and Tables

**Figure 1 cancers-16-00274-f001:**
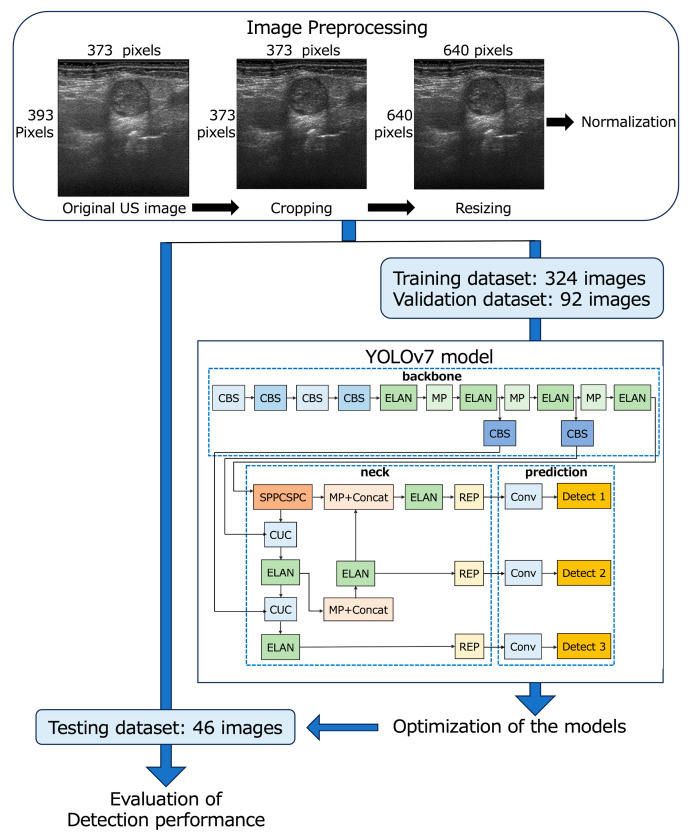
Flowchart of the development of YOLOv7-based models.

**Figure 2 cancers-16-00274-f002:**
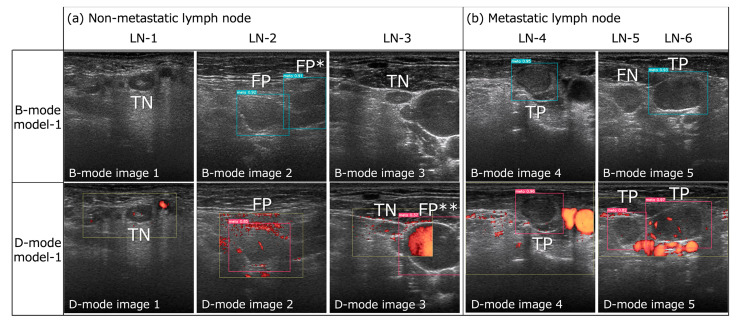
Successful and unsuccessful examples of metastatic lymph node detection by B- and D-mode model-1 (confidence score of 0.1 for both) in B- and D-mode model images, respectively. (**a**) Both B- and D-mode model-1 exhibited true negatives (TN) for LN-1 and -3 and false positives (FP) for LN-2. However, B-mode model-1 registered a false positive (FP*) for the cross-section of the sternocleidomastoid muscle in B-mode image 2, and D-mode model-1 registered a false positive (FP**) for the internal jugular vein in D-mode image 3 (this was the only FP for the non-LN area by D-mode model-1). (**b**) Both B- and D-mode model-1 demonstrated true positives (TP) for LN-4 and -6, but B-mode model-1 exhibited a false negative (FN) for LN-5, while D-mode model-1 achieved a TP result.

**Figure 3 cancers-16-00274-f003:**
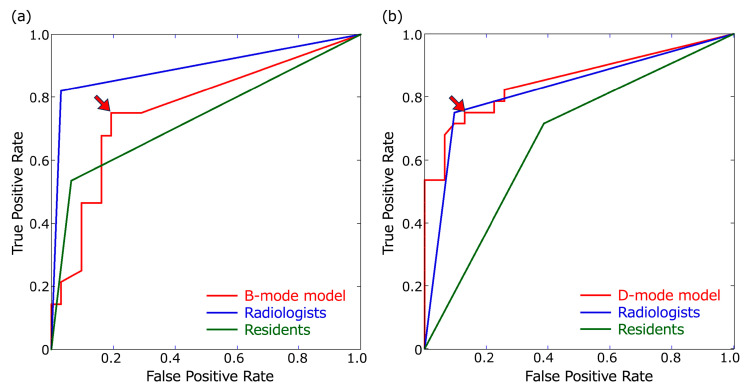
Receiver operating characteristic (ROC) curves. ROC curves of the B- and D-mode models were based on confidence scores of ≥0.1. (**a**) ROC curves for B-mode images. The threshold of the B-mode model for obtaining the largest area under the ROC curve (AUC) was a confidence score of 0.23 (arrowhead). (**b**) ROC curves for the D-mode images. The threshold of the D-mode model for obtaining the largest AUC was a confidence score of 0.48 (arrowhead).

**Table 1 cancers-16-00274-t001:** Number of images and LNs for B-mode and D-mode training, validation, and testing datasets.

	Training Datasets	Validation Datasets	Testing Datasets	Total
No. of images	324	92	46	462
No. of LNs	365	116	59	540
No. of metastatic LNs	183	50	28	261
No. of non-metastatic LNs	182	66	31	279

**Table 2 cancers-16-00274-t002:** Evaluation indicators for the detection performance of metastatic LNs.

			Detection Result	
			Detected as Metastatic LNs	Not Detected as Metastatic LNs
Pathology	Metastatic LNs	N (A + B) = 28	True positive: A	False negative: B
	Non-metastatic LNs	N (C + D) = 31	False positive: C	True negative: D
Non-LN areas detected as metastatic LNs			False positive for non-LN area: E	

A–E indicate the number (N) of LNs or non-LN areas incorrectly detected as LNs. The evaluation indicators of detection performance for metastatic LNs were determined using the following equation: recall = A/28, precision = A/(A + C + E); F1-score = 2 × precision × recall/(precision + recall); false-positive rate for non-LN area (= false positives for non-LN area in one test image) = E/46; accuracy = (A + D)/59.

**Table 3 cancers-16-00274-t003:** Detection performance for metastatic LNs.

**B-mode images**	**B-mode model-1**	**B-mode model-2**	**Radiologists**	**Residents**
Recall	0.75	0.75	0.821	0.536
Precision	0.618	0.724	0.958	0.517
F1-score	0.677	0.737	0.885	0.526
False-positive rate for non-LN area	0.087	0.043	0	0.261
Accuracy	0.729	0.78	0.898	0.746
AUC (95% CI)	0.73 (0.601–0.829)	0.778 (0.652–0.868)	0.895 (0.786–0.951)	0.736 (0.62–0.826)
**D-mode images**	**D-mode model-1**	**D-mode model-2**	**Radiologists**	**Residents**
Recall	0.821	0.75	0.75	0.714
Precision	0.719	0.84	0.875	0.606
F1-score	0.767	0.792	0.808	0.656
False-positive rate for non-LN area	0.022	0	0	0.022
Accuracy	0.78	0.814	0.831	0.661
AUC (95% CI)	0.782 (0.657–0.870)	0.81 (0.689–0.892)	0.827 (0.707–0.904)	0.664 (0.533–0.773)
**Comparison of accuracy, recall, and AUC between B-mode and D-mode images**				
*p* value for recall *	0.687	1	0.687	0.063
*p* value for accuracy *	0.549	0.727	0.289	0.359
*p* value for AUC **	0.365	0.518	0.169	0.270

*p*-values by * binomial test and ** DeLong test.

**Table 4 cancers-16-00274-t004:** Comparison of recall, accuracy, and AUC between the models and observers.

**B-mode images**	B-mode model-1		B-mode model-2		Radiologists		Residents	
B-mode model-1		Recall	1		0.727		0.146	
	Accuracy/AUC							
B-mode model-2	0.250/0.073			Recall	0.727		0.146	
			Accuracy/AUC					
Radiologists	0.031/0.0166		0.118/0.0747			Recall	0.008	
					Accuracy/AUC			
Residents	1/0.9351		0.804/0.5242		0.012/0.0022			Recall
							Accuracy/AUC	
**D-mode images**	D-mode model-1		D-mode model-2		Radiologists		Residents	
D-mode model-1		Recall	0.5		0.727		0.508	
	Accuracy/AUC							
D-mode model-2	0.687/0.4645			Recall	1		1	
			Accuracy/AUC					
Radiologists	0.607/0.4914		1/0.7951			Recall	1	
					Accuracy/AUC			
Residents	0.230/0.1591		0.093/0.0615		0.064/0.0419			Recall
							Accuracy/AUC	

The numbers show the *p*-values obtained using the DeLong test for AUC and the binominal test for recall and accuracy. Numbers in bold indicate statistical significance.

**Table 5 cancers-16-00274-t005:** Recall at each cervical level.

**B-mode images**	**B-mode model-1**	**B-mode model-2**	**Radiologists**	**Residents**
Level I	0.909	0.909	0.909	0.364
Level II	0.8	0.8	0.9	0.7
Level III + IV	0.429	0.429	0.571	0.571
**D-mode images**	**D-mode model-1**	**D-mode model-2**	**Radiologists**	**Residents**
Level I	0.727	0.636	0.727	0.636
Level II	1	1	1	0.7
Level III + IV	0.714	0.571	0.429	0.857

**Table 6 cancers-16-00274-t006:** Agreement in the identification of metastatic LNs between the models and observers.

B-Mode Images	Kappa Value	D-Mode Images	Kappa Value
B-mode model-1 vs. B-mode model-2	0.898	D-mode model-1 vs. D-mode model-2	0.798
B-mode model-1 vs. Radiologists	0.392	D-mode model-1 vs. Radiologists	0.496
B-mode model-1 vs. Residents	0.361	D-mode model-1 vs. Residents	0.149
B-mode model-2 vs. Radiologists	0.483	D-mode model-2 vs. Radiologists	0.546
B-mode mode-2 vs. Residents	0.437	D-mode model-2 vs. Residents	0.230
Radiologists vs. Residents	0.595	Radiologists vs. Residents	0.199

The numbers indicate Cohen’s kappa values.

## Data Availability

The data that support the findings of this study are available from the corresponding author, M.S. (Misa Sumi), upon reasonable request.

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
