# Peer review of "Metastatic Lymph Node Detection on Ultrasound Images Using YOLOv7 in Patients with Head and Neck Squamous Cell Carcinoma"

_cancers, 2024, doi:10.3390/cancers16020274_

Round 1

Reviewer 1 Report

Comments and Suggestions for Authors

Dear Authors,

I congratulate you on this very complex study.

However, there are some aspects that require your attention.

There are many abbreviations in the manuscript and you need to include a list at the end of the article.

In the discussion section you need to expand the differential diagnosis of lumps in the neck. Because such pathology can bias the AI result in the future clinical settings. There is still a possibility for an AI software to consider as False Positive images of lymphangiomas, or thyroglossal duct cysts for example. Reference this differential diagnosis to articles such as https://doi.org/10.11152/mu-1422 or  https://doi.org/10.11152/mu.2013.2066.173.aco

In Table 6 correct the line with B-mode mode-2 vs. Residents

Clearly the use of AI is the future but you need to be a little bit precautious. At this stage of development such AI applications should be limited only to screening. You could underline the fact that this will help in reducing the pressure in already crowded imaging departments.

Looking forward to receiving the improved version of your manuscript.

Author Response

Thank you very much for taking the time to review this manuscript. We feel your comments have helped us significantly improve the paper. Please find the detailed responses below and the corresponding revisions/corrections in yellow in the re-submitted files.

Reviewer 2 Report

Comments and Suggestions for Authors

The manuscript addresses the timely issue of the application of deep learning to diagnostic imaging, with a focus on metastatic lymph node detection. The study is well-written and data clearly presented. I would suggest the authors to add a post-hoc sample size estimation and to perform a power analysis to further corroborate their results.

Author Response

Thank you very much for taking the time to review this manuscript. Thank you for your thoughtful and valuable review.

Reviewer 3 Report

Comments and Suggestions for Authors

The study focuses on improving the diagnosis of cervical lymph node metastasis in HNSCC, a significant factor affecting patient prognosis. The research involves developing deep learning models using YOLOv7 for identifying metastatic lymph nodes in ultrasound images.

1. Introduction:

The literature review could be more comprehensive. The paper briefly mentions various imaging modalities but does not provide enough context on why these modalities have limitations and how the proposed method addresses these gaps.

The description of ultrasound diagnostics is informative, but the transition to discussing DL applications in this field is somewhat abrupt. More background on the challenges of conventional ultrasound diagnostics methods and how DL can overcome these challenges would strengthen the argument for the study's necessity.

A more detailed justification for selecting YOLOv7, including its advantages over other DL models in this specific application, would be beneficial.

Other than real-life data, GAN and CFD simulation will be another good approach to enlarge the dataset. I would recommend citing the following papers: https://doi.org/10.1115/1.4053651; https://doi.org/10.1038/s41598-021-89636-z.

2. Materials and Methods:

More technical details on how the images were processed or standardized before being fed into the deep learning model would be useful. 

The detailed structure of YOLO V7 and any modifications for transfer learning should be included.

The method used for bounding box annotations and the validation of these annotations (to ensure accuracy and consistency) is not mentioned.

The comparison with radiologists and residents is a strong aspect, but the calibration process using 20 images needs more explanation. Clarifying how these images were selected and ensuring they are representative of the broader dataset is essential to avoid bias.

3. Results:

AUC curves are needed.

The comparison between the models (B-mode and D-mode) and the human experts (radiologists and residents) is well-conducted. However, a more detailed discussion on why certain metrics are higher for one group over the other could provide deeper insights into the strengths and limitations of the models versus human expertise.

It would be helpful to delve deeper into why certain false positives occurred and what this implies about the model's limitations or areas where it could be improved.

Discussing why the agreement levels vary and how this might impact the clinical application of the models would add depth to the analysis.

Model interpretability analysis should be included.

Author Response

Thank you for taking time to review this manuscript. We feel your comments have helped us significantly improve the paper. Please find the detailed responses below and the corresponding revisions/corrections in yellow in the re-submitted files.

Round 2

Reviewer 3 Report

Comments and Suggestions for Authors The manuscript has been sufficiently improved to warrant publication in Cancers